# Eye Movement Control in Tibetan Reading: The Roles of Word Length and Frequency

**DOI:** 10.3390/brainsci12091205

**Published:** 2022-09-07

**Authors:** Xiao-Wei Li, Shan Li, Lei Gao, Zi-Bei Niu, Dan-Hui Wang, Man Zeng, Tian-Zhi Li, Xue-Jun Bai, Xiao-Lei Gao

**Affiliations:** 1Plateau Brain Science Research Center, Tibet University, Lhasa 850000, China; 2Education Department, Lhasa Normal College, Lhasa 850007, China; 3Key Research Base of Humanities and Social Sciences, Institute of Psychology and Behavior, Tianjin Normal University, Tianjin 300000, China

**Keywords:** Tibetan reading, word length, word frequency, eye movement control

## Abstract

We investigated the effects of word length and frequency on eye movement control during Tibetan reading through two experiments. A preliminary experiment examined the predictive effect of word length and frequency on fixation duration and landing position using multiple linear regression analysis. In the formal experiment, we manipulated the length and frequency of target words simultaneously to investigate the effects of word length and frequency on fixation duration and landing position in Tibetan reading. In this study, we found that: (1) there were significant word-length and word-frequency effects affecting all lexical processing in Tibetan reading; (2) there are preferred viewing locations in Tibetan reading; specifically, for short words, it is the end, whereas for long words, it spans from the center to the beginning of the word; (3) word frequency does not affect preferred viewing location in Tibetan reading; (4) the preferred viewing position and the interaction of word length and viewing position found in this study supported the “strategy-tactics” approach.

## 1. Introduction

Reading is an important cognitive activity and an essential means to acquire knowledge and understand external information. During reading, the eye movement system must control when and where to move the eyes in real time. These aspects introduce two fundamental questions in the study of eye movement control, that is, which mechanisms drive “when” and “where” decisions? Fixation duration addresses the question of when readers move their eyes. Measures related to fixation duration include those related to characters or words and those related to phrases or sentences, such as single fixation duration, total fixation duration, etc. [1]. Landing position examines the question of where to move the eyes [2]. The landing position effect refers to the tendency of the reader’s saccades to land on a particular word position during reading [3,4]. The landing position effect performs differently in alphabetic writing and Chinese. In alphabetic writing, there are two landing position effects. One is the optimal viewing position (OVP), which is the fastest position to recognize the word, located at the center of the word [5]. It has been found that the nearer the initial fixation lands from the OVP, the lower the probability of refixation on the word, showing a “U” curve. In particular, the lowest probability of re-fixation on the word occurs when readers’ initial fixation lands on the OVP of the word [5,6,7]. The second is the preferred viewing location (PVL), where the readers’ initial fixation lands between the word’s center and beginning (i.e., about 1/4 of the word) [8]. In Chinese, for OVP, different from alphabetic writing, it has been found that when readers’ initial fixation lands on the beginning of the word, the probability of re-fixation on the word is highest, while the probability of re-fixation on the word is lowest when it lands on the word end [9]. Additionally, for PVL, Yang and McConkie together with Tsai and McConkie found that participants’ initial fixation on two-character words were evenly distributed over each character, and the distribution of landing position on all target words was very similar, showing a smooth curve with no PVL [10,11]. However, Yan et al. strictly controlled the ambiguity of word boundary and found the PVL effect landed on the word center, showing a convex curve [9,12].

Word length and frequency affecting eye movement control are essential factors during reading [2,13]. There is a close relationship between word length and frequency. First, word frequency may decrease with the increase in length. Second, word frequency modulates word-length effects. Third, the word-length effect is more prominent in low- than in high-frequency words [14]. In Chinese and alphabetic writing, many studies have shown that word length affects the fixation duration and there is a robust word-length effect. The word-length effect is such that, compared with long words, short words are easier to recognize for readers, which leads to shorter fixation duration and less fixation counts [15]. This is because readers obtain the most lexical information about the word when they fixate on it, since the word is presented in their foveal region (2 degrees in the center of their vision). The foveal region is the area of highest visual acuity. From the parafoveal region (extending from the foveal region to about 5 degrees on either side of fixation) to the peripheral region (everything beyond the parafoveal region), the visual acuity decreases as the distance from the foveal region increases, and the reader obtains less and less information [1,16,17]. Therefore, compared with long words, short words occupy a closer visual space and are more concentrated in the foveal region when they are fixated on by readers, who may need only one fixation to complete the recognition of the word, resulting in a faster processing of short words than long words. Thus, the word-length effect occurs. For example, Ren et al. used single-character words as a baseline condition to explore the difference in fixation duration of two-, three-, and four-character vs. single-character words [18]. The results revealed a significant word-length effect on total and first fixation duration and gaze duration. Li et al. also found that readers required longer gaze and total fixation duration to process long words compared with short words [19]. In addition, Rayner et al. found a significant word-length effect on fixation duration in English reading [20]. Bricolo et al. selected Italian words composed of four, six, or eight letters, and found that word length affected gaze duration in Italian reading [21]. With the increase in word length, the gaze duration also increased.

Furthermore, many studies have shown that word length also affects the landing position. Tong et al. used single-character and two-character words as experimental materials to explore the effect of word length on landing position in Chinese reading [22]. The results indicated that word length affected readers’ landing position in Chinese reading. The landing position of long words was farther from the beginning of the word than that of short words. Research on the effect of word length on landing position in alphabetic writing shows that the landing position of long words tended to be left of the word’s center, whereas that of short words was more toward the word’s end [23,24].

According to above, word length is an important factor affecting eye movement control during reading. What effect does word frequency, an attribute closely related to word length, have on eye movement control? A great number of studies have found that word frequency affects fixation duration and there is a significant word-frequency effect in Chinese and alphabetic writing [25,26,27,28]. In reading, the reader mainly obtains information from the foveal region, but the role of the peripheral region in information acquisition is also important [29,30,31,32,33,34,35]. Readers are able to acquire certain information from the parafoveal region through preview processing, leading to an increase in reading efficiency, which is the preview benefit [2,36,37]. It has been found that the foveal processing load affects the amount of preview benefit readers obtain from the parafoveal region [2,38]. Word frequency is one of the measures of the foveal processing load [39]. Compared to low-frequency words, readers are more familiar with high-frequency words and have less processing load, which leads to the word-frequency effect.

Regarding the influence of word frequency on landing position, some studies have found that the frequency of the first letter of a word affects landing position [40,41]. When the first letter of the word has low frequency, the participants initially fixate on the beginning of the word. Other studies have found that word frequency did not affect landing position. Using the boundary paradigm, Angele and Rayner manipulated the frequency of the target word and found that word frequency did not affect landing position during reading [42]. Wu et al. adopted a single-factor within-subject design [43]. The independent variable contained two levels (i.e., word frequency: low and high), and the results confirmed that word frequency did not affect landing position. The researchers suggested that landing position might relate to word segmentation. Successful word segmentation helps readers land their next saccade on the center of the target word, thus increasing the speed of subsequent lexical processing. In contrast, word frequency did not affect landing position, possibly because the difference in the degree of pre-processing between the two did not reach a level affecting subsequent saccade target selection.

China is a multi-ethnic, multi-lingual country. In addition to the Han, there are 55 ethnic minorities having more than 80 languages belonging to multiple language families. Tibetan is one of the ancient national languages, belonging to the Tibeto-Burman language family of the Sino-Tibetan language family. It has alphabetic writing with 30 consonants, 4 vowel symbols, and 5 reverse letters as its basic character/word units. Tibetan is an alphabetic script composed of letters. Its written structure shows the features of linear development. It has syllable boundary markers (syllable separators). Tibetan’s structure is similar to that of the Chinese language. It is written around a “base character” appended before and after and written up and down, displaying a certain stereoscopic quality [35]. Therefore, Tibetan is a unique language, including both alphabetic and ideographic writing characteristics.

During the Tibetan reading process, a question arises as to whether the effects of word length and frequency on fixation duration and landing position will show similarities with alphabetic writing because Tibetan is an alphabetic script. Perhaps some similarities with Chinese exist because Tibetan has Chinese features. On the other hand, it may display uniqueness, as it combines the elements of alphabetic writing and Chinese. Unfortunately, research in this area is limited, and the underlying mechanisms remain unclear. Therefore, in our preliminary experiment with Tibetan college students as participants, using eye movement tracking technology, we investigated the effects of word length and frequency on eye movement control in Tibetan reading through multiple linear regression analysis.

## 2. Preliminary Experiment

### 2.1. Materials and Methods

#### 2.1.1. Participants

Forty-two students at Tibet University (20 females; mean age = 20.50 years, *SD* = 1.13) participated in the experiment. The participants were all native Tibetan speakers and had normal or corrected-to-normal vision. They signed an informed consent form before taking part in the study.

#### 2.1.2. Materials

We selected sentences from upper elementary school Tibetan textbooks and comparable level extracurricular books. They were adapted appropriately, forming 79 Tibetan sentences. Sentence length was between 19 and 33 characters, and there was no semantic and syntactic ambiguity in the sentences. Figure 1 shows an example of the experimental materials.

Fifteen Tibetan college students were asked to evaluate the naturalness of sentences on a 7-point scale (1 = entirely unnatural, 7 = entirely natural), and another 15 Tibetan college students evaluated the difficulty of sentences on a 5-point scale (1 = very easy to understand, 5 = very difficult to understand). According to the evaluation results, 60 sentences were finally selected as formal experimental sentences (naturalness: *M* = 5.81, *SD* = 0.36; difficulty: *M* = 1.51, *SD* = 0.16), indicating that the experimental materials were natural and easy, which met the experimental requirements. We excluded students who participated in the material evaluation from the experiment.

#### 2.1.3. Apparatus

Participants’ eye movements were recorded using an SR Research Ltd. EyeLink 1000 Plus eye-tracking system (Ottawa, ON, Canada) with a sampling rate of 1000 Hz. Sentences were displayed on a 21-inch CRT monitor (SONY MuLtiscanG520 from the Sony Group Corporation, Tokyo, Japan; resolution: 1024 × 768 pixels; refresh rate: 140 Hz). Stimuli were presented in Microsoft Himalaya font size 32, and each Tibetan character was about 15 pixels wide on the monitor. Participants were seated about 65 cm from the monitor. Each Tibetan character corresponded to approximately 0.6 degrees of visual angle at this viewing distance.

#### 2.1.4. Procedure

Each subject was tested individually. When participants arrived at the laboratory, we first familiarized them with the environment. Then, they were given experimental instructions and a brief description of the apparatus. Participants were seated in a designated location. We instructed them to minimize head movements as much as possible during the experiment to ensure the accuracy of the data. To ensure that the eye tracker accurately recorded the participants’ eye movement trajectory, they completed a three-point calibration and validation procedure until they attained an average error below 0.25 degrees [35]. Recalibration and revalidation were conducted when necessary (i.e., when error increased beyond 0.25 degrees). Eight practice sentences were set to ensure that participants were familiar with the procedure. In addition, we set 20 question sentences to ensure that participants read carefully and understood the content they read. The entire experiment lasted approximately 20 min.

#### 2.1.5. Measures

With reference to previous studies [44,45], we computed the following eye movement measures for each two-syllable target word region in the sentences: (a) single fixation duration (SFD, the duration of fixations when only one fixation was made during first pass reading); (b) first fixation duration (FFD, the duration of the first fixation on a region during first pass reading); (c) gaze duration (GD, the sum of all fixations on a region from first entering the region until leaving it during first pass reading); (d) total fixation duration (TFD, the sum of all fixations on a region); and (e) landing position (LP, the position where reader fixate, not only as a landing site of the previous saccade but also as the launch site of the next saccade).

### 2.2. Results

Multiple linear regression analysis was conducted on the eye movement data using R (Version 4.1.1) [46]. The mean comprehension accuracy for all participants was 80.41%. With reference to previous literature [47], the data were excluded according to the following criteria: (1) trials with track-loss or error (due to the participants’ head movements and other factors in the experiment); (2) short (<80 ms) and long (>1200 ms) fixations; (3) trials for sentences receiving fewer than five fixations; and (4) outliers beyond three standard deviations from the mean. In total, we removed 1.23% of the trials.

The preliminary experiment aimed to investigate the effects of word length and frequency on fixation duration and landing position. The first step in data processing was determining the length and frequency of words in experimental sentences. The second step was to perform a regression analysis of word length and frequency on eye movement measures. Word length was determined according to the number of characters (*M* = 2.02, *SD* = 0.77). We consulted the Modern Tibetan Frequency Dictionary [48] to determine word frequency (per ten million; *M* = 43, *SD* = 62).

#### 2.2.1. Multiple Linear Regression Analysis of Word Length and Word Frequency on Eye Movement Measures

Table 1 shows the significance test results of word length and frequency on eye movement measures.

Results related to SFD indicate that word length and frequency were significantly negatively correlated with single fixation duration. The longer length and higher frequency related to shorter single fixation duration (multiple R^2^ = 0.32%, adjusted R^2^ = 0.30%), indicating that 0.30% of the variation in single fixation duration can be explained by the linear relationship between word length and frequency, and showing a poorly fitted model. We found that word length and frequency significantly negatively correlated with FFD. The longer the length and the higher the frequency, the shorter the first fixation duration (multiple R^2^ = 0.30%, adjusted R^2^ = 0.28%), indicating that 0.28% of the variation in first fixation duration is explained by a linear relationship between word length and frequency, and also showing a poorly fitted model. Word length significantly positively correlated with gaze duration. Gaze duration increased with greater word length. Word frequency was significantly negatively correlated with gaze duration; the higher the frequency, the shorter the gaze duration (multiple R^2^ = 0.66%, adjusted R^2^ = 0.64%). This indicates that 0.64% of the variation in gaze duration can be explained through the linear relationship between word length and frequency, showing a poorly fitted model. For TFD, word length significantly positively correlated with total fixation duration. The longer the word length, the longer the total fixation duration. In contrast, word frequency significantly negatively correlated with total fixation duration. Total fixation duration decreased with higher word frequency (multiple R^2^ = 1.40%, adjusted R^2^ = 1.37%); this indicates that 1.37% of the variation in total fixation duration can be explained using a linear relationship between word length and word frequency, and shows a poorly fitted model. Finally, concerning LP, word length significantly positively correlated with landing position (multiple R^2^ = 15.26%, adjusted R^2^ = 15.25%), indicating that 15.25% of the variation in landing position is explained through a linear relationship between word length and frequency.

The test revealed that word length and word frequency may jointly predict landing position. There was a significant effect of word length on landing position.

#### 2.2.2. Analysis of Landing Position Distribution

To further analyze the influence of word length on landing position in Tibetan reading, referring to previous literature [45], we performed three measures, including initial landing position distribution, initial landing position distribution in single-fixation cases, and initial landing position distribution in multiple-fixation cases.

Initial landing position distribution

To investigate the initial landing position distribution in Tibetan reading, we analyzed the percentage of fixations in the area of interest. Figure 2 displays the results.

Initial landing position distribution in single-fixation cases

The first fixation includes both single- and multiple-fixation cases. In single-fixation, the reader has only one fixation on the word. To investigate the initial landing position distribution in single-fixation cases during Tibetan reading, we analyzed the percentage of single-fixation in the area of interest. Figure 3 shows the results.

The results showed that when there was only one fixation, the initial landing position moved from the end to the center of the word with the increase in word length.

Initial landing position distribution in multiple-fixation cases

In multiple-fixation, the reader has two or more fixations on the word. To investigate the initial landing position distribution in multiple-fixation cases during Tibetan reading, we analyzed the percentage of multiple-fixation in the area of interest. Figure 4 displays the results.

The results showed that when word length was 2–7 characters, the initial landing position occurred more toward the beginning of the word in multiple-fixation cases. Compared with the initial landing position in single-fixation cases, it was closer to the word’s beginning. Thus, readers’ initial landing position shifted to the beginning of the word in multiple-fixation cases.

Word length had an important influence on fixation duration in Tibetan reading. With the increase in word length, total fixation and gaze durations were longer. In contrast, first and single fixation durations were shorter. The result was consistent with the research of Zang et al. [49]. The higher word frequency relates to a shorter fixation duration. In addition, word length significantly affects landing position in Tibetan reading. Specifically, with the increase in the word length, the initial landing position in single-fixation cases moved from the end to the center of the word. This finding was inconsistent with previous research of Chinese and alphabetic script reading. In single-fixation instances, the initial landing position occurred more toward the center of the word during Chinese reading. In contrast, with the increase in word length in alphabetic script reading, the initial landing position moved from the end to the center and beginning of the word. In multiple-fixation cases, our result was consistent with previous research on Chinese and alphabetic script reading showing that the initial landing position moved toward the beginning of the word [9,24,50].

The preliminary experiment results showed that word length and frequency in Tibetan reading predicted fixation duration and landing position. However, the preliminary experiment applied multiple linear regression analysis and did not control attributes of target words (i.e., word length and frequency). Therefore, we proceeded with the formal experiment to investigate further the effects of word and word frequency on fixation duration and landing position by controlling the length and frequency of target words. Based on previous research [18,20,35], we proposed (1) that there are significant word-length and frequency effects in Tibetan reading; that is, readers have longer fixation durations on longer and lower-frequency words. Based on the fact that Tibetan is an alphabetic script, we hypothesize (2) that word length affects the preferred viewing location in Tibetan reading. The preferred viewing location of short words tends toward their ends, whereas it moves from the center to the beginning for long words. Based on preliminary experiment results, we proposed the hypothesis (3) that word frequency does not affect preferred viewing location in Tibetan reading.

## 3. Formal Experiment

### 3.1. Materials and Methods

#### 3.1.1. Participants

Sixty-four students at Tibet University (33 females; Mean Age = 20.25 years, *SD* = 1.08) participated in the experiment. The participants were all native Tibetan speakers and had normal or corrected-to-normal vision. They signed an informed consent form before taking part in the experiment.

#### 3.1.2. Design

A 2 (word length: long words, short words) × 2 (word frequency: high frequency, low frequency) within-subjects experimental design was used.

#### 3.1.3. Materials

Selection of experimental materials

First, we initially determined that high-frequency words occurred more than 200 times per 10 million, whereas low-frequency words occurred less than 50 times per 10 million. Second, we selected 48 pairs of high- and low-frequency words to form two-syllable words of different lengths using the Modern Tibetan Frequency Dictionary [48]. The length of short words was 2–4 characters, and the length of long words was 5–7 characters. Finally, four groups of two-syllable words were formed into target words: (1) long words with high frequency (HL); (2) short words with high frequency (HS); (3) long words with low frequency (LL); and (4) short words with low frequency (LS). There were 48 target words in each group, and all the target words were nouns.

Evaluation of experimental materials (including vocabulary and sentence)

##### Vocabulary Evaluation

We tested the word length and frequency of the selected four groups of target words for differences. Table 2 shows means and standard deviations for the word length and frequency across four target word groups.

There was a significant main effect for word length (*F* (3, 141) = 151.64, *p* < 0.001, $\eta_{p}^{2}$ = 0.76, 95%CI = (3.24, 5.55)). Post hoc multiple comparison tests (LSD) showed that there was no significant difference between the word length of the target word in the HL and LL conditions (*p* > 0.05). There was no significant difference between the length of the target word in the HS and LS conditions (*p* > 0.05). There were significant differences between other conditions (*p* < 0.05). There was a significant main effect for word frequency (*F* (3, 141) = 47.01, *p* < 0.001, $\eta_{p}^{2}$ = 0.50, 95%CI = (0.02, 0.37)). Multiple comparison testing (LSD) showed that there was no significant difference between the frequency of the target word in the HL and HS conditions (*p* > 0.05). There were significant differences between other conditions (*p* < 0.05).

##### Sentence Evaluation

A total of 192 Tibetan sentences (48 sentence frames) were compiled to ensure that all parts of the sentence frame were completely consistent, excluding the 192 target words determined by vocabulary evaluation results. The sentence length was between 15 and 40 characters. There were no semantic and syntactic ambiguities in the sentences. Figure 5 shows an example of the experimental materials. Thirty Tibetan college students were asked to evaluate the difficulty and naturalness of the sentences on a 5-point scale (1 = very easy to understand/entirely unnatural, 5 = very difficult to understand/ entirely natural; difficulty: *M* = 1.91, *SD* = 0.21; naturalness: *M* = 4.07, *SD* = 0.29). In addition, 30 students were given the first part of the experimental sentence up to and including the character to the left of the target word. They were asked to provide the next word in the sentence (i.e., predict the target word). The predictability of the target words was 7%. The evaluation results showed that the Tibetan experimental sentences were easy, natural, and not predictive, which met the experimental requirements. Students who participated in the vocabulary and sentence evaluation were excluded from the formal experiment.

The 192 Tibetan sentences that met the experimental requirements were divided into four blocks using a Latin square balance design. Each block contained four conditions of equal number (i.e., each block had 48 sentences, including 12 sentences for each condition). In addition, before the formal experiment, we added eight practice sentences and eight question sentences to each block to ensure that the participants read carefully. The participants were asked to answer “yes” or “no” by pressing a button. Each participant was required to read only one block. The experiment was completed in approximately 20 min. Figure 5 displays an example of the experimental materials.

#### 3.1.4. Apparatus and Procedure

These were identical to the preliminary experiment.

### 3.2. Results

The mean comprehension accuracy for all participants was 91.99%. The criteria for excluding data were identical to those of the preliminary experiment. In total, 5.50% of the trials were removed.

As in previous studies [45,51,52,53,54], we selected eye movement measures of temporal and spatial dimensions. Temporal dimension measures were identical to those in the preliminary experiment. We selected spatial dimension measures including: (a) average forward saccadic amplitude (AFSA, i.e., the mean saccadic length of all forward fixations from left to right); (b) skipping rate (SR, i.e., the probability that the area of interest is skipped during the first pass reading); (c) the average initial landing position (AILP, i.e., the first fixation on the target word but regardless of the total number of fixations there are on that word) and its distribution; (d) the average initial landing position and its distribution in single-fixation cases (i.e., the reader has only one fixation on the word); (e) the average initial landing position and its distribution in multiple-fixation cases (i.e., the reader has two or more fixations on the word); and (f) regression rate (RR. i.e., the probability of regression from the area of interest to the left after fixation).

Linear Mixed Models (LMMs) were created using the lme4 package [55] in R (Version 4.1.1) [46] for data analysis. Word length and frequency were treated as fixed factors. We first ran a model with a maximal random effects structure consisting of slopes for all fixed effects across subjects and items for each eye movement measure. However, this process was trimmed for models that failed to converge, starting with items and then participants, removing the correlations between factors, interactions, and random slopes until the model converged. LMMs were able to integrate all the raw data into the model for statistical analysis, greatly improving the utilization of data. We reported regression coefficients, standard errors, and *t*- or *z*-values, with *t/z* > 1.96 meaning *p* < 0.05. With reference to previous references [52,56,57], we performed log transformation of fixation duration data (i.e., four temporal dimension eye movement measures) to increase the normality, and skipping data were analyzed using logistic GLMMs given the binary nature of the variable.

#### 3.2.1. The Temporal Dimension Eye Movement Measures Results

Table 3 shows descriptive statistical results for temporal dimension eye movement measures.

Table 4 displays results of *b*, *SE*, and *t* value for each temporal dimension eye movement measure.

For first fixation duration (FFD), the main effect of word length was significant. The FFD of short words was shorter than that of long words. The main effect of word frequency was significant. FFD for low-frequency words was longer than that of high-frequency words. There was no significant interaction between word length and frequency.

For single fixation duration (SFD), gaze duration (GD) and total fixation duration (TFD) both showed similar patterns of effects with FFD, that is, there were significant main effects of word length, significant main effects of word frequency, and there was no significant interaction between word length and word frequency. Specifically, on SFD, GD, and TFD, short words are significantly shorter than long words, whereas low-frequency words are significantly longer than high-frequency words. Supported by the early (first fixation duration, single fixation duration, and gaze duration), and late (total fixation duration) measures, the above results suggested a significant word-length effect and word-frequency effect in Tibetan reading, both of which respectively affected lexical processing.

#### 3.2.2. The Spatial Dimension Eye Movement Measures Results

Table 5 shows descriptive statistical results of the spatial dimension eye movement measures.

Table 6 displays results of *b*, *SE*, and *t/z* value for each spatial dimension eye movement measure.

For regression rate (RR), the main effect of word length was significant, and the RR of short words was higher than that of long words. The main effect of word frequency was significant, and the RR of low-frequency words was higher than that of high-frequency words. There was a significant interaction between word length and frequency.

Both skipping rate (SR) and the average forward saccadic amplitude (AFSA) showed similar patterns of effects. The main effect of word frequency was not significant. The main effect of word length was significant, the SR of short words was higher than that of long words, and the AFSA for short words was larger than that of long words. There was no significant interaction between word length and frequency. Average initial landing position (AILP), AILP in single-fixation cases, and AILP in multiple-fixation cases showed similar patterns of effects. The main effects of word length were significant, with the AILP of long words more to the right than that of short words. The main effect of word frequency was not significant. There was no significant interaction between word length and frequency. For average initial landing position (AILP) and its distribution, we performed repeated measure analysis of variance for the percentage of fixation landing in each region. Figure 6 shows the distribution of initial landing position across the four conditions.

The results showed that the main effect of word length was significant in regions 2 and 4 (*F_s_* (1, 63) = 4.89, *p* = 0.03, $\eta_{p}^{2}$ = 0.07; *F_s_* (1, 63) = 19.31, *p* < 0.001, $\eta_{p}^{2}$ = 0.24, 95%CI = (0.07, 0.08)); multiple comparison (LSD) found that participants made more fixations when the target words were short. In region 7, the main effect of word frequency was significant, *F_s_* (1, 63) = 8.00, *p* = 0.006, $\eta_{p}^{2}$ = 0.11; multiple comparison (LSD) found that participants made more first fixations when the target words were with high frequency. In other regions, the main effect of word length and word frequency, and the interaction between word length and word frequency, were not significant (*Fs.* < 1, *p* > 0.05).

In brief, the results of AILP and its distribution showed that when the target words were long, the participants’ AILP landed closer the word’s beginning. Conversely, when target words were short, participants’ fixations landed between the center and end of the word.

For AILP and its distribution in single-fixation cases, we performed repeated-measures analysis of variance for the percentage of fixation landing within each region in single-fixation cases. Figure 7 shows the distribution of initial landing position across the four conditions.

The results showed that the main effect of word length was significant in regions 3 and 4 (*F_s_* (1, 63) = 38.71, *p* < 0.001, $\eta_{p}^{2}$ = 0.38, 95%CI = (0.07, 0.08); *F_s_* (1, 63) = 25.09, *p* < 0.001, $\eta_{p}^{2}$ = 0.29, 95%CI = (0.07, 0.08)). Multiple comparisons (LSD) found that participants made more fixations when target words were short. In Region 7, the main effect of word frequency was significant (*F_s_* (1, 63) = 8.13, *p* = 0.006, $\eta_{p}^{2}$ = 0.110). Multiple comparison tests (LSD) found that participants made more first fixations when target words were high frequency. In Region 4, the interaction between word length and word was significant (*F_s_* (1, 63) = 4.36, *p* = 0.04, $\eta_{p}^{2}$ = 0.07). Simple effect analysis found that participants in the low-frequency condition had more fixations on short than on long words. In other regions, the main effect of word length and frequency, and interaction between word length and frequency, were not significant (*Fs.* < 1, *p* > 0.05).

In brief, AILP results in single-fixation cases and its distribution showed that when target words were long, participants’ fixations landed more often between the center and the beginning of the word. When the target words were short, the participants’ AILP landed more toward the word’s end.

For AILP and its distribution in multiple-fixation cases, we performed repeated-measures analysis of variance for the percentage of fixation landing within each region in multiple-fixation cases. Figure 8 displays the distribution of initial landing position across the four conditions.

The results showed that the main effect of word length was significant in regions 3 and 4 (*F_s_* (1, 63) = 15.45, *p* < 0.001, $\eta_{p}^{2}$ = 0.20, 95%CI = (0.03, 0.06); *F_s_* (1, 63) = 6.79, *p* = 0.01, $\eta_{p}^{2}$ = 0.10). Multiple comparison tests (LSD) found, regarding word length, participants made more fixations when target words were short. In Regions 1, 2, and 4, the main effect of word frequency was significant (*F_s_* (1, 63) = 13.84, *p* < 0.001, $\eta_{p}^{2}$ = 0.18, 95%CI = (0.05, 0.09); *F_s_* (1, 63) = 36.98, *p* < 0.001, $\eta_{p}^{2}$ = 0.37, 95%CI = (0.03, 0.09); *F_s_* (1, 63) = 4.37, *p* = 0.04, $\eta_{p}^{2}$ = 0.07). Multiple comparison tests (LSD) found participants made more fixations when the target words were high frequency. In Regions 2 and 4, the interaction between word length and frequency was significant, (*F_s_* (1, 63) = 24.35, *p* < 0.001, $\eta_{p}^{2}$ = 0.28, 95%CI = (0.02, 0.10); *F_s_* (1, 63) = 16.71, *p* < 0.001, $\eta_{p}^{2}$ = 0.21, 95%CI = (0.02, 0.06)). Simple effect analysis found that participants in the low-frequency condition had more fixations on short words than on long words. In other regions, the main effect of word length and frequency, and the interaction between word length and frequency, were not significant, *Fs.* < 1, *p* > 0.05.

Briefly stated, the AILP results in multiple-fixation cases and its distribution showed that, regardless of word length, the participants’ landing position occurred more toward the word’s beginning.

Our results show a preferred viewing location in Tibetan reading. Word length affected the preferred viewing location. For short words, this location was the end; for long words, it was from the center to the beginning of the word. Word frequency did not affect the preferred viewing location. This finding was consistent with previous studies on alphabetic writing [23,24].

## 4. Discussion

There are two aspects to eye movement control in reading: the mechanisms that determine when readers move their eyes, and the mechanisms that determine where to move them. In other words, we need to identify factors affecting fixation duration and landing position. Based on previous studies on Chinese and alphabetic writing, word length and frequency are two important factors affecting eye movement control during reading. However, for Tibetan, a unique language belonging to the Sino-Tibetan language family featuring alphabetic writing, there is far less available research concerning the effects of word length and word frequency on eye movement control in Tibetan reading.

Therefore, the present study used eye movement tracking technology to investigate the effects of word length and frequency on eye movement control in Tibetan reading. The preliminary experiment investigated the predictive effects of word length and frequency on fixation duration and landing position during Tibetan reading. The formal experiment further examined the effects of word length and frequency on fixation duration and landing position in Tibetan reading by manipulating the target words’ length and frequency.

Results of the preliminary experiment showed that word length negatively correlated with first fixation duration (FFD) and single fixation duration (SFD). One possible reason for this is that some attributes of a word n + 1 (e.g., word length or frequency) cause SFD and FFD of word n to decrease with expanding word length [32], that is, the reverse parafoveal-on-foveal effect. This indicates the difficulty of word n + 1 processing is inversely proportional to the processing duration of word n. The more difficult the processing of word n + 1, the shorter the processing duration of word n [58]. Second, SFD and FFD reflect an earlier processing stage of words, whereas word length shows an opposite trend in very early measures. Word length information of Tibetan words may not be immediately available at the early stage of visual and lexical processing. Although gaze duration (GD) also reflects the early processing stage of words, compared with FFD, it also reflects the integration process of content. Total fixation duration (TFD) is also a late measure, related to word integration into sentences [37]. Word length significantly positively correlated with GD and TFD, suggesting that the word-length effect is a product of refixation. Word frequency significantly negatively correlated with measures of fixation duration. With higher frequency, the fixation durations on the words were shorter, indicating that there is a significant word-frequency effect in Tibetan reading. This result was consistent with those of existing studies [35].

Furthermore, preliminary experiment results on the distribution of landing position showed that, with increasing word length, the landing position moved from the end to the center of the word in the single-fixation cases. This finding was inconsistent with the results of Chinese and alphabetic script reading studies. Additionally, in the multiple-fixation cases, the landing position was the beginning of the word, consistent with previous studies [24,59,60]. In this case, it may depend on whether the number of times the reader fixates on the word is planned or spontaneous. When the readers’ first fixation lands on the center of the word (i.e., the optimal viewing position), they need only fixate on the word once to recognize it. However, since there is a degree of error in saccades, the readers’ first fixation may deviate from the optimal viewing position, i.e., landing on the word beginning. The readers then must re-fixate to recognize the word.

The formal experiment results showed word length and frequency effects in Tibetan reading, consistent with the results of the existing studies [20,35,61,62]. The E-Z reader model might explain this result. This model suggests that readers launch eye movements for word n + 1 only after completing the early familiarity test for word n. Subsequently they continue to complete word recognition for word n until they recognize it, shifting attention to word n + 1 to begin the process again. Visual acuity, highest in the fovea, influences the familiarity test. As word length increases, visual acuity gradually decreases along with processing efficiency, which requires more fixations to complete word recognition. Thus, as word length increases, fixation duration will correspondingly be longer. Similarly, word frequency affects the familiarity test. Therefore, fixation duration for high-frequency words is shorter than low-frequency words. Secondly, it is because there is a robust word-length effect and word-frequency effect in Chinese and alphabetic writing. In this study, word-length and word-frequency effects were found in Tibetan reading, further supporting the validity of the robust quality of word-length and word-frequency effects.

The results of formal experiment also showed that there was a preferred viewing location in Tibetan reading. Word length affects readers’ preferred viewing location, consistent with other research on English and other alphabetic writing [24]. The result may reflect a “strategy-tactics” approach used by Tibetan undergraduates during saccades [5]. The “strategy-tactics” approach suggests that readers adopt an inter-word saccade strategy during the process of reading. This method guides the reader to fixate on the optimal viewing position for each target word. However, this strategy has a degree of risk. Some saccades land outside the optimal viewing position. Consequently, readers may use within-word tactics to compensate. Thus, if a saccade does not land on the optimal viewing position, it re-fixates on the target word. This approach ensures that each target word will be fixated from the optimal viewing position in single-fixation cases, and from two different position in multiple-fixation cases.

Although this study achieved the expected research purpose, the following shortcomings still exist: (1) as a unique ethnic language, the question of whether other special factors (e.g., word predictability, semantic transparency) affect eye movement control during Tibetan reading need to be further studied; (2) the cognitive processing of eye movement control in Tibetan reading can be further analyzed in combination with brain imaging techniques.

In conclusion, during Tibetan reading, there is a significant word-length and word-frequency effect. In addition, there is a preferred viewing location, mainly affected by low-level factors. In contrast, high-level factors do not affect landing position, consistent with previous research on alphabetic writing [63]. Tibetan and Chinese belong to the Sino-Tibetan language family and have similar stereoscopic writing structures. Chinese is ideographic, while Tibetan is alphabetic, showing similar characteristics to alphabetic writing in other language types, including pronunciation transparency and inter-character/word markers. Therefore, in Tibetan reading, the effect of word length on the preferred viewing location is similar to that in alphabetic writing.

## 5. Conclusions

(1) There were significant word-length and word-frequency effects in Tibetan reading, affected by and reflected in lexical processing.

(2) There was a preferred viewing location in Tibetan reading. Word length affected the preferred viewing location in Tibetan reading. With increasing word length, the preferred viewing location moved from the end toward the center and then to the beginning of the word.

(3) Word frequency had no effect on the preferred viewing location in Tibetan reading.

(4) The findings regarding effects of word length on preferred viewing location supported the “strategy-tactics” approach.

## Figures and Tables

**Figure 1 brainsci-12-01205-f001:**
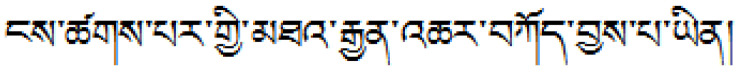
An example of the materials used in the preliminary experiment. The meaning of the sentence is “I designed a newspaper frame”.

**Figure 2 brainsci-12-01205-f002:**
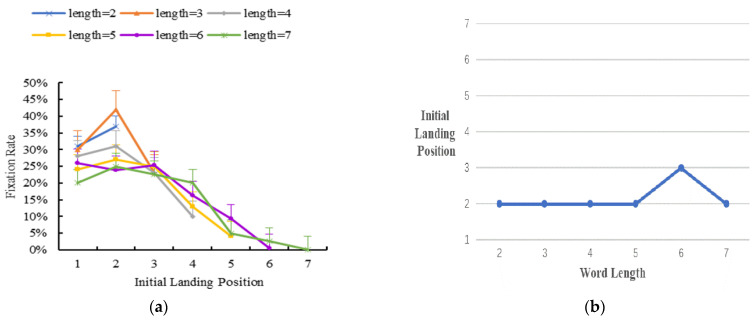
(**a**) The distribution of initial landing positions. Error bars show the standard error of the mean. (**b**) A further illustration of (**a**), and the y axis is the most probably initial landing position for trials. The results showed that the initial landing position was more toward the word end when the word length was two characters. When word length was three characters, it landed more toward the word center. Finally, when word length was 4–7 characters, it landed more toward the word beginning. Thus, the distribution peak of readers’ initial landing position varied with word length. Specifically, with increased word length, readers’ initial landing position moved from the end to the center and the beginning of the word.

**Figure 3 brainsci-12-01205-f003:**
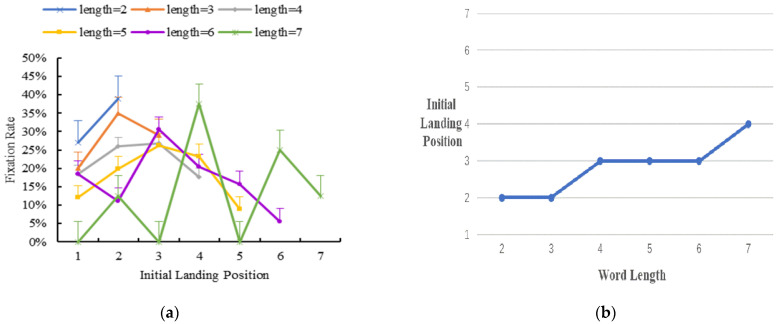
(**a**) The distribution of initial landing position in single-fixation cases. Error bars show the standard error of the mean. (**b**) A further illustration of (**a**), and the y axis is the most probably initial landing position for trials in single-fixation cases.

**Figure 4 brainsci-12-01205-f004:**
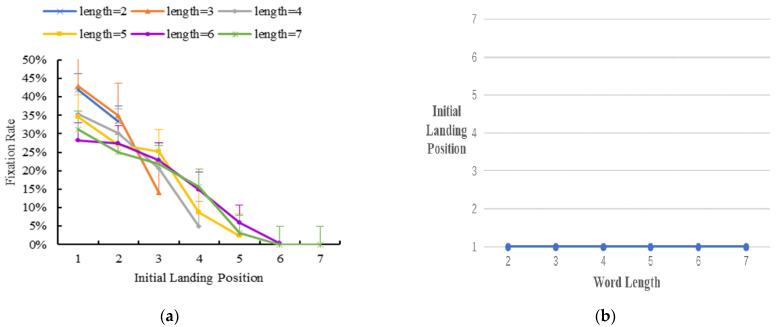
(**a**) The distribution of initial landing position in multiple-fixation cases. Error bars show the standard error of the mean. (**b**) A further illustration of (**a**), and the y axis is the most probably initial landing position for trials in multiple-fixation cases.

**Figure 5 brainsci-12-01205-f005:**
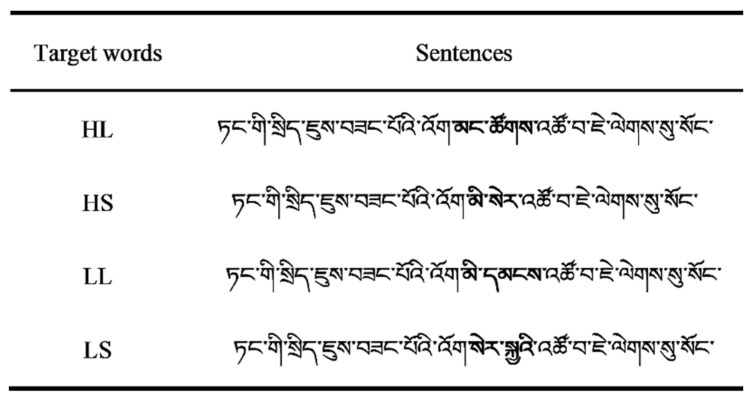
An example of materials used in the formal experiment. The target words are in bold, but were presented as regular font in the experiment. HL: long words with high frequency; HS: short words with high frequency; LL: long words with low frequency; LS: short words with low frequency. The translated sentence is “The **people**’s life is getting better and better under the Party’s policy”.

**Figure 6 brainsci-12-01205-f006:**
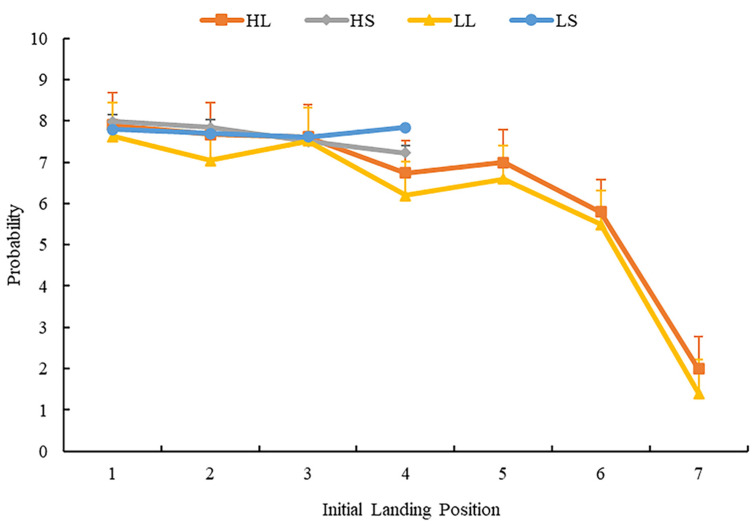
The distribution of average initial landing position. Error bars show the standard error of the mean.

**Figure 7 brainsci-12-01205-f007:**
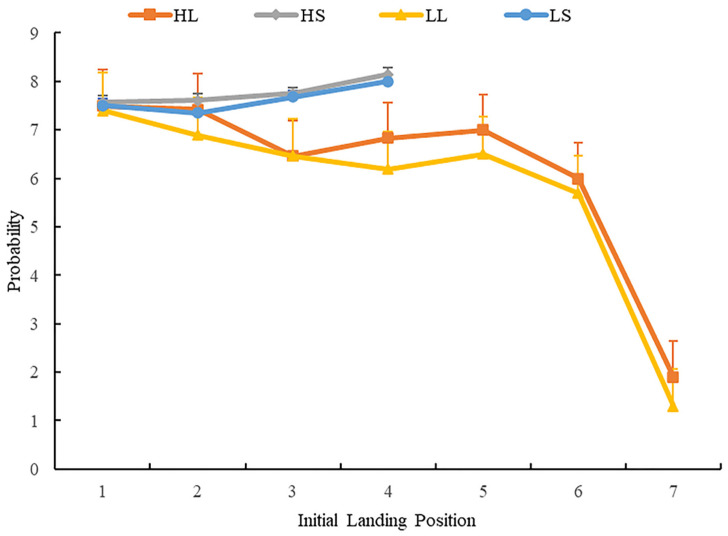
The distribution of average initial landing position in single-fixation cases. Error bars show the standard error of the mean.

**Figure 8 brainsci-12-01205-f008:**
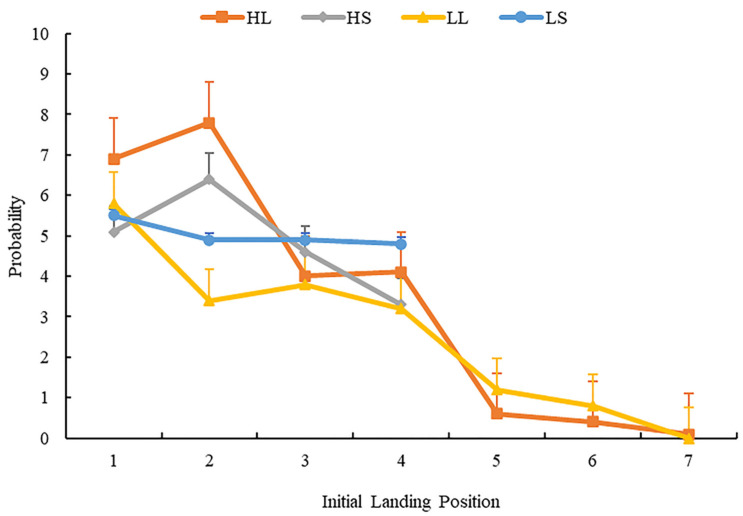
The distribution of average initial landing position in multiple-fixation cases. Error bars show the standard error of the mean.

**Table 1 brainsci-12-01205-t001:** Significance test results of word length and word frequency on eye movement measures in the preliminary experiment.

Measures	Word Length			Word Frequency			Multiple R-Squared	Adjusted R-Squared
	Estimate	*t* Value	Pr (>|*t*|)	Estimate	*t* Value	Pr (>|*t*|)		
SFD	−5.81	−4.05	<0.001 ***	−0.80	−4.20	<0.001 ***	0.0032	0.0030
FFD	−6.40	−4.80	<0.001 ***	−0.62	−3.39	<0.001 ***	0.0030	0.0028
GD	11.35	6.34	<0.001 ***	−0.64	−2.61	<0.001 ***	0.0066	0.0064
TFD	29.96	8.42	<0.001 ***	−1.94	−4.49	<0.001 ***	0.0140	0.0137
LP	7.05	39.50	<0.001 ***	0.05	1.90	0.06	0.1526	0.1525

SFD = single fixation duration; FFD = first fixation duration; GD = gaze duration; TFD = total fixation duration; LP = landing position. Estimate means the regression coefficient estimates; Pr(>|*t*|) means *p* value; Multiple R-squared means coefficient of determination in multiple linear regression, i.e., R^2^; Adjusted R-squared means adjusted R^2^. *** *p* < 0.001.

**Table 2 brainsci-12-01205-t002:** Means and standard deviations for word length and word frequency across four target word groups.

Target Words	Word Length (the Number of Characters)	Word Frequency (Per Ten Million)
HL	5.29 (0.50)	248 (202)
HS	3.46 (0.62)	299 (233)
LL	5.37 (0.61)	28 (26)
LS	3.44 (0.68)	39 (31)

HL: long words with high frequency; HS: short words with high frequency; LL: long words with low frequency; LS: short words with low frequency. Standard deviations are provided in parentheses.

**Table 3 brainsci-12-01205-t003:** Means and standard deviations for temporal dimension eye movement measures.

Target Words	FFD	SFD	GD	TFD
HL	248.24 (2.47)	249.08 (2.50)	254.38 (2.64)	320.61 (4.89)
HS	232.57 (2.33)	232.01 (2.30)	239.31 (2.62)	289.34 (4.24)
LL	275.61 (3.34)	275.72 (3.39)	287.64 (3.66)	393.15 (6.71)
LS	255.44 (2.39)	253.08 (2.50)	262.75 (2.57)	322.33 (4.23)

Standard deviations are provided in parentheses. HL: long words with high frequency; HS: short words with high frequency; LL: long words with low frequency; LS: short words with low frequency. SFD = single fixation duration; FFD = first fixation duration; GD = gaze duration; TFD = total fixation duration. All measures are in milliseconds.

**Table 4 brainsci-12-01205-t004:** Summary of statistical effects for temporal dimension eye movement measures.

Measures	Fixed Effects	*b*	*SE*	*t*
FFD	Word length	−0.004	0.010	−4.29 ***
	Word frequency	0.005	0.009	5.18 ***
	Word length × Word frequency	0.001	0.020	0.27
SFD	Word length	−0.043	0.010	−4.31 **
	Word frequency	0.048	0.010	4.80 ***
	Word length × Word frequency	0.012	0.020	0.59
GD	Word length	−0.004	0.001	−4.17 ***
	Word frequency	0.005	0.001	4.95 ***
	Word length × Word frequency	−0.0003	0.002	−0.02
TFD	Word length	−0.007	0.001	−5.27 ***
	Word frequency	0.005	0.001	3.90 ***
	Word length × Word frequency	−0.0005	0.003	−0.18

FFD = first fixation duration; SFD = single fixation duration; GD = gaze duration; TFD = total fixation duration. *b* = regression coefficient. ** *p* < 0.01, *** *p* < 0.001.

**Table 5 brainsci-12-01205-t005:** Means and standard deviations for spatial dimension eye movement measures.

Target Words	RR	SR	AFSA	AILP	AILP in Single-Fixation Cases	AILP in Multiple-Fixation Cases
HL	0.25 (0.084)	0.13 (0.008)	2.14 (0.032)	0.78 (0.011)	0.77 (0.011)	0.89 (0.043)
HS	0.27 (0.078)	0.15 (0.009)	3.31 (0.044)	0.74 (0.010)	0.73 (0.011)	0.68 (0.062)
LL	0.27 (0.095)	0.13 (0.008)	2.03 (0.027)	0.79 (0.012)	0.77 (0.012)	0.94 (0.050)
LS	0.28 (0.121)	0.16 (0.009)	3.42 (0.044)	0.71 (0.010)	0.70 (0.010)	0.74 (0.051)

Standard deviations are provided in parentheses. HL: long words with high frequency; HS: short words with high frequency; LL: long words with low frequency; LS: short words with low frequency. RR = regression rate; SR = skipping rate; AFSA = average forward saccadic amplitude; AILP = the average initial landing position.

**Table 6 brainsci-12-01205-t006:** Summary of statistical effects for spatial dimension eye movement measures.

Measures	Fixed Effects	*b*	*SE*	*t/z*
RR	Word length	0.05	0.01	4.58 ***
	Word frequency	0.03	0.01	3.72 **
	Word length × Word frequency	−0.09	0.02	−4.39 ***
SR	Word length	0.20	0.07	2.65 **
	Word frequency	0.03	0.07	0.41
	Word length × Word frequency	0.18	0.14	1.23
AFSA	Word length	0.52	0.04	13.72 ***
	Word frequency	−0.01	0.03	−0.28
	Word length × Word frequency	0.07	0.08	0.90
AILP	Word length	−0.10	0.03	−3.20 ***
	Word frequency	−0.04	0.08	−0.45
	Word length × Word frequency	0.01	0.06	0.11
AILP in single-fixation cases	Word length	−0.08	0.03	−2.62 ***
	Word frequency	−0.04	0.04	−1.34
	Word length × Word frequency	−0.02	0.03	−0.39
AILP in multiple-fixation cases	Word length	−0.40	0.13	−3.16 **
	Word frequency	−0.02	0.12	−0.18
	Word length × Word frequency	0.13	0.25	0.53

RR = regression rate; SR = skipping rate; AFSA = average forward saccadic amplitude; AILP = the average initial landing position. *b* = regression coefficient. ** *p* < 0.01, *** *p* < 0.001.

## Data Availability

Data materials can be obtained by contacting the corresponding author.

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
