# Peer review of "Eye Movement Control in Tibetan Reading: The Roles of Word Length and Frequency"

_brainsci, 2022, doi:10.3390/brainsci12091205_

Round 1
Reviewer 1 Report
It is really great to see eye movement research being conducted in other languages and writing systems. Far too much of eye movement research is conducted in English or using the Latin alphabet. The second experiment presents valuable information about eye movement behavior in Tibetan based on the effects of word frequency and word length. I would like to see this research published as it is valuable in broadening eye movement research globally; however, as you will see from my comments below, there are some major issues throughout the paper to be improved. I hope the comments are helpful to the authors and will help you improve the quality of the paper for future publication.
Page 1
In the opening paragraph, the authors discuss landing position as a measure of answering the question where to move the eyes. This is true, but it is only one of many measures that can be used to answer this question. The authors should consider saccade distance and regressive eye movements as other measures related to the where question.
Page 2
The authors state “A regression or re-fixation may occur if the readers’ initial fixation lands on the preferred viewing location.”
I am unfamiliar with this effect, and so a citation should be included. Even if there is a direct citation for this statement, it is a bit misleading. It implies that landing on the preferred viewing location is of primary importance for causing a regression or refixation, but there are many other reasons for either of these eye movements, which are all far more impactful than landing on the preferred viewing location.
In other words, it’s odd to cite landing on a preferred viewing location as a cause of regressions or refixations when other factors are more likely to cause these eye movements.
“In Chinese and alphabetic writing reading,…”
I don’t think the word reading needs to be included in this sentence
The first main paragraph on page 2 cites many pieces of evidence to support the word length effect. Instead of listing all these effects individually, it could be more helpful for the reader if you explain why the word length effect happens. You could discuss visual acuity and parafoveal processing for a more complete explanation of the effect.
To say that “some studies have found a significant word frequency effect…” is an understatement. This is a very robust effect and that should be clear. The statement saying “some” implies that some have not found this effect, but I am not aware of any such credible evidence.
On the Liversedge study, you said that they “controlled the frequency of target words” and found a word frequency effect, but that would be impossible. You cannot find an effect on a variable if that variable is controlled. Word frequency must have varied for reading times to vary by word frequency.
Again, in this paragraph many pieces of evidence are cited to note the word frequency effect, but there is no explanation for WHY this effect happens.
Page 3
Is there any reason that the materials had to be rated as easy? Does this present any challenges for generalizability of the results? Does this limit the words that could be used in the study?
Page 4
Though it might not be central to your research question, including the rates for skipping, regression in, and regression out are likely valuable for other researchers. Since there is such limited research in Tibetan, the more measures you include, the better.
More distributional information about your materials must be included beyond measures of central tendency. What was the range of frequency and word length, how many words of each length did you have, how many words were considered high or low frequency (typically below 30 counts per million is considered low frequency)? It is customary to report frequency in counts per million (not counts per ten million), so it would be good to convert those numbers.
Pages 5
It is not clear why linear regression was used in this study instead of linear mixed effects models that were used in the second study. The authors must know about the benefits of LME over linear regression and why eye movement researchers do not use linear regression or multiple RM ANOVAs anymore. The authors should reanalyze the data using LME as they did in the second study.
Table 1
Scientific notation is not necessary for p values. You can simply put < .001
Page 8
The plus or minus symbol in Table 2 is odd because plus or minus is not the same thing as standard deviation. This is a misleading symbol for what the numbers truly mean. You could simply put the SD in parentheses next to the M.
Page 10
Thanks for including all the details about the LMM models. This looks good. There is one thing I’m confused on – the logistic transformation for saccade data. The log transformation for reading time data is correct, but I’ve never seen a logistic transformation for saccade data. Perhaps this is just my own naivete. Could you provide a citation for this or explain why you used it? I’m not seeing this as a common practice.
Page 11
This paragraph could be written far more concisely without repeating the same thing over and over. Simply put, there were significant effects of frequency and length across all four reading time measure with no interaction. I don’t think a paragraph with the model values included is necessary since those are already in the table.
The same applies to page 12 where there was only a significant effect of length but not frequency or interaction across all measures. This can really simplify things for the reader and make it easier to process all of the information.
The only other issues I’m observing with the data is that it appears as though length and frequency were analyzed as categorical binary variables. In reality, these are continuous variables with a full range of frequencies and lengths. It is always better to analyze continuous variables as continuous rather than categorical. There are many papers on this as a bad practice (see here: https://scholar.google.com/scholar?hl=en&as_sdt=0%2C22&q=dichotomizing+continuous+variables&btnG=)
Page 15, Lines 521-523
This sentence seems to imply that a parafoveal on foveal effect is the cause of the reverse word length effect. These effects are extremely controversial with not much support. If you think this is the case here, you could analyze this effect to provide support for your conclusion. This is a testable claim.
The claim that visual information may not be available for visual or lexical processing is odd. Why? What is the basis for such a claim? Why would this be different in Tibetan compared to other alphabetic languages? Word length is often the earliest piece of information to be extracted.
It seems as though there is a lot of grasping to explain this effect without much support. Further, a later paragraph on the same page contradicts these conclusions when discussing the second experiment.
Finally, given the value of this paper as its first investigation into Tibetan eye movements, it would be greatly beneficial for the academic community to make the data publicly available.
Author Response
Please see the attachment.
Thank you for your work!

Reviewer 2 Report
The manuscript by Xiao-wei Li et al., ‘Eye movement control in Tibetan reading: The roles of word length and frequency’, showed that word length and frequency affect eye movement in Tibetan reading. The manuscript also revealed some interesting related details, such as ‘there are preferred viewing locations in Tibetan reading, specifically, for short words, it is the end, while for long words, it spans from the center to the beginning of the word’, ‘the preferred viewing position and the interaction of word length and viewing position found in this study supported the “strategy-tactics” approach’, etc.
In sum, I feel the study was well performed and their findings are interesting. Most of the conclusions are solid. Most of the analyses, e.g., statistical analysis, are well done. However, there are a few major concerns regarding the credibility of one of their main statements described below:
Major concerns:
For Figure 2, the authors claimed that “The results showed that the initial landing position was more toward the word end when the word length was two characters” (line 234). However, there is no related statistical test nor did the authors show individual data points, leading their conclusion ill supported. Please at least show that this statement is statistically significant, as is done in Figure 6 and 7. Please at least add the s.e.m. of each data point. Please also do this for Figures 3-8. Perhaps also include individual data points, maybe in the supplements, to give the readers some concrete feelings and confidence about the variation of the relationship.
The authors stated that ‘The results showed that when there was only one fixation, the initial landing position moved from the end to the center of the word with the increase of word length’ (Line 247). However, there is no plot directly showing this correlation. Readers will have to feel and guess by themselves using data from Figure 2. To make this point, the authors should add one more plot, where the x axis is word length, ranges from 2 to 7, and the y axis is the average (or most probably) initial landing position for trials with only one fixation. The correlation between the x and y axis needs to be positive and statistically significant.
Minor concerns:
1. In Line 21, the authors say that ‘The results suggested that…’. If all of their main conclusions can, at best, only be suggested by their data, perhaps the work need further improvement or only publish as a theoretical paper. I think this is just an issue with English writing, since I believe that the authors showed enough data for points (2) and (3) (conditioned on the statistical tests mentioned in my Major concerns being significant).
2. In Line 21: ‘there were significant word length and frequency effects affecting all lexical processing’. Is such a general statement novel? Or did your study cover all lexical processing? If not, please use a weaker tone or be more specific, such as ‘consistent with other studies [citations]’, or add ‘in Tibetan reading’.
3. In Line 222: ‘The test revealed that word length and word frequency might jointly predict landing position. There was a significant effect of word length on landing position.’ However, the P value between word frequency and Landing position is as large as 0.06, suggesting it is not playing a role in predicting landing position. Please state this sentence differently, such as delete ‘word frequency’.
4. I only begin to realize the significance of this work after reading the super long introduction. I believe the authors could make this process much shorter and less miserable by adjusting their writing. In the current Abstract, there is no description about why this study is interesting nor any problems existed in the current research field. Please add some to attract more readers. The authors can also cut their Introduction a bit.
5. In Tables 1 and 2, consider using full phrases instead of acronyms. I have to jump back and forth between the tables and legends many times to understand the content of the Tables.
6. In Figure 2, it is hard to distinguish different lines, let alone to see the maximum point of each line, which is closely related to one of the authors’ main statements. Maybe try plot different curves separately.
Author Response

(The authors gave the same response as above.)

Round 2
Reviewer 1 Report
Thank you for addressing most of my comments. Most of this was done well, but there are still some remaining issues that I’ll explain below:
The response to Point 8 basically says no, but why not? It can only improve your paper to include more eye movement measures. Word skipping and regression rates are some of the most commonly reported measures and provides a lot of valuable information about processing. Given that this is the first paper to investigate Tibetan reading, you should provide as much information as possible for future researchers. You certainly have these measure available, so they should be reported.
If you choose to maintain a no answer in response to this request, there should be at least some justification for not including these measures.
The response to Point 9 does not answer the question of how the categories were created. If you have high and low frequency words, there must have been a criterion for deciding what is considered high or low frequency. Why did you consider one word high frequency and another low frequency? Likewise, if you have long and short words, there must have been a criterion for deciding how many characters is short or long. Simply reporting the mean does not provide enough information about the range of words in each of these categories. This information is important to report. You note that there is no standardized rule for determining high and low frequency in Tibetan, but you must have used a rule to decide which of your words were high or low frequency. What was it?
The response to Point 10 is most inadequate of all. There is no real explanation provided for why the wrong type of analysis was used. Multiple regression is simply the wrong analysis to use for eye movement and reading research. The fact that this study is exploratory does not justify using the wrong type of analysis. LMM should be used for both exploratory and non-exploratory analyses. There is no legitimate justification provided for continuing to use linear regression. The correct analysis is used in Experiment 2 and should also be used in Experiment 1.
The response to Point 16 is still describing a parafoveal-on-foveal effect. A parafoveal on foveal effect is when characteristics of word n+1 have an effect on the processing of word n. You said “the more difficult word n+1 processing is, the shorter the processing duration of word n becomes.” That is literally a parafoveal on foveal effect. My previous comment about the controversy surrounding these effects remains and must be discussed further. In your comment, you cite a couple sources for evidence of parafoveal on foveal effects, but that is exactly the controversy. Some have found it and many have not. This is no simple topic in eye movement research and when one is found it warrants much further discussion.
Point 18 is not addressed at all. Given the contradictions that are noted, this comment cannot simply be ignored.
Author Response
Please see the attachment.
Thank you again for your work!

Reviewer 2 Report
I agree with most of the authors' reply and feel that this revised version gets improved a lot, especially on data presentation. I'm OK with this paper being published and only have two more minor follow-up comments:
1. Since you have added the s.e.m. to each data point in your figures, please specify them in the figure legends.
2. This is a further elaboration respect to my original point 2 since the authors stated that they did not fully get what I meant. In current Line 294, you said 'The results showed that when there was only one fixation, the initial landing position moved from the end to the center of the word with the increase of word length'. In order to understand this conclusion that you wrote, a reader has to go to Figure 3, manually calculate the average (or most likely) initial landing position for each word-length scenario, put these dots together in their brain (or on a piece of paper), and then see if the trend is true or not. What I meant is that if you can make an additional plot to show this trend in a straightforward way, that would be great.
Author Response

(The authors gave the same response as above.)
